# Examination of the High Tensile Ductility Improvement in an As-Solutionized AA7075 Alloy with the Aid of a Friction Stir Process

**Ming-Hsiang Ku, Fei-Yi Hung**  **and Truan-Sheng Lui \***

Department of Materials Science and Engineering, National Cheng Kung University, Tainan 701, Taiwan; n58051049@mail.ncku.edu.tw (M.-H.K.); fyhung@mail.ncku.edu.tw (F.-Y.H.)

**\*** Correspondence: luits@mail.ncku.edu.tw; Tel./Fax: +886-6-2757575 (ext. 62931)

**Abstract:** The ductility enhancement of an AA7075 aluminum alloy aided by a friction stir process (FSP) and various heat treatments was investigated and compared in terms of outcome with full annealing (O). The results indicate that a big improvement in the tensile ductility was achieved by freezing the sample at temperatures below 0 °C after the solution treatment and water quenching (W treatment), and further improvement could be acquired via a friction stir process due to grain refinement (<6 µm). Thus, the observed improvement in tensile ductility can be explained by the fact that the W treatment and friction stir processing scheme had an increased strain-hardening effect and decreased the presence of intermetallic particles that are harmful to uniform tensile deformation, consequently causing strain localization in the early stage of tensile deformation, which suggests that these treatment are a potential solution for insufficient formability. In general, the elongation to failure values for the W and FSP-treated specimens (>40%) were at least 1.5-fold greater than that of the annealed alloy. In addition, serrated flow could be observed in the tensile flow curves, and both the Piobert–Lüders effect and the Portevin–LeChatelier (PL) effect could be observed. The enhancement in the tensile ductility was examined in terms of the existence of intermetallic particles and the supersaturated concentration of the solid solution.

**Keywords:** AA7075 aluminum alloy; friction stir process (FSP); heat treatment; tensile ductility; Portevin–LeChatelier effect

## 1. Introduction

Among the hot rolled and hot extruded aluminum alloy thick plates (>2 mm), AA7075 is generally regarded as an ultra-high strength alloy, which is widely used for structural applications [1–3]. The mechanical response of this alloy with various post-heat treatments (PHTs) when subjected to straining is an investigation field of wide interest. Full annealing [4–6] normally has been employed to improve ductility due to mitigating the necking phenomenon under tensile deformation. However, disappointingly, it still exhibits comparatively low ductility owing to insufficient strain hardening, which is the main drawback limiting its practical application.

From a practical viewpoint, the strain hardening response associated with strain localization under tension needs to be clarified in order to, subsequently, develop an optimum post-heat treatment condition. In the latter metal-working process, it is well known that tensile strain, when acting at the plate surface, can lead to macroscopic failure due to insufficient tensile ductility. Many previous investigations in the literature [6–8] have suggested that the tensile properties in the post-heat-treated state are highly dependent on microstructural features. It is, therefore, crucial to examine the tensile deformation behavior of AA7075 aluminum alloys, which are capable of absorbing tensile strain

during deformation and are usually associated with enhancing tensile ductility both in terms of the stress–strain response and in terms of fracture features.

Previous reports [7,8] indicated that aluminum alloys such as Al-Zn-Mg-(Cu) and 2024 aluminum alloy gained improved ductility by solution heat treatment and subsequent rapid water quenching (W treatment). Their tensile curve exhibited intense yielding, corresponding to the Portevin–LeChatelier (PL) effect after W heat treatment [9–16]. This PL effect is a tensile deformation characteristic in many aluminum alloys. When the PL effect occurs, the stress–strain curve of tensile deformation exhibits a serrated flow phenomenon. On the other hand, an alloy that undergoes uniform deformation indicates the strain-hardening capacity and determines the amount of plastic deformation that can be accommodated prior to the onset of strain localization. Therefore, an increment in the strain hardening rate can mitigate strain localization and promote uniform elongation of 7075 alloys.

For precipitation hardening of high-strength AA7075 aluminum alloys, various precipitates play an important role during different aging stages in the main strengthening mechanism. Only a secondary role is attributed to grain size and deformation-hardening character. It is well known that these precipitation hardening alloys, when acting on severe tensile deformation processes, often lead to macroscopic failure due to insufficient tensile ductility.

The friction stir process (FSP) [17] is a well-known microstructural modification method due to its capacity to reduce defects and improve tensile ductility. FSP uses a rotating tool to generate heat while moving along the substrate. FSP results in severe plastic deformation. Microstructural observation has shown that the process area consists of uniformly fine dynamic recrystallization grains, and the grain/subgrain size in the stir zone develops through a continuous dynamic recrystallization (CDRX) mechanism. Recently, the application was extended to a wider area by multipass friction stir processing [18–21] as an aid-forming process. Numerous studies [22–24] have investigated the metallurgical factors and the related mechanical properties of friction stir processed materials. Our previous investigations [22,25,26] suggested that the tensile ductility can be improved for many aluminum alloys through the development of a controlled cooling method for the purpose of acquiring grain refinement, phase transformation, or a supersaturated solid solution. However, this research focused on the effect of a rapid quenching treatment combined with a friction stir process on the ability to achieve higher tensile ductility. Efforts were directed toward understanding the deformation kinetics of alloys, which have been predominantly attributed to the solution effects associated with the presence of intermetallic particles in the matrix.

## 2. Materials and Methods

In this study, we used commercial hot rolled AA7075-T651 aluminum plate with a thickness of 5 mm. Its chemical composition (in wt %) is Al-5.7Zn-2.4Mg-1.7Cu-0.3Cr-0.2Fe. The specimens went through a solution treatment and T6 treatment to become a 7075-T6 aluminum alloy as the base metal (BM).

Figure 1 shows a schematic illustration of the FSP. The employed FSP tool had a pin diameter of 6 mm, a shoulder diameter of 18 mm, and a pin length of 3.3 mm. The processing direction (PD) was parallel to the rolling direction (RD). We used a 440 rpm rotational speed, while the tool moving speed was fixed at 0.58 mm s$^{-1}$ with a 1.5° tool angle, and a downward pushing force of 38.7 MPa. The FSPed specimens received two heat treatments: (1) natural aging (NA: 40 °C, 100 h), and (2) solution treatment and water quenching (W: 480 °C, 1 h). The specimen codes for all experiments are listed in Table 1.

**Table 1.** Experimental conditions and specimen codes.

| Base Metal | Rotational Speed (rpm) | AF [a] | NA [b] | W [c] | O [d] |
|---|---|---|---|---|---|
| BM-T6 | - | - | - | BM-W | BM-O |
| | 440 | 440-AF | 440-NA | 440-W | - |

[a] AF: As fabricated. [b] NA: natural aging (40 °C, 100 h). [c] W: solution treatment (480 °C, 1 h) + quenching (cool water) + placed in dry ice. [d] O: Heating (415 °C, 2 h) + Furnace cooling (20 °C/h) to 260 °C + heating (260 °C, 6 h) + Furnace cooling to room temperature (RT).

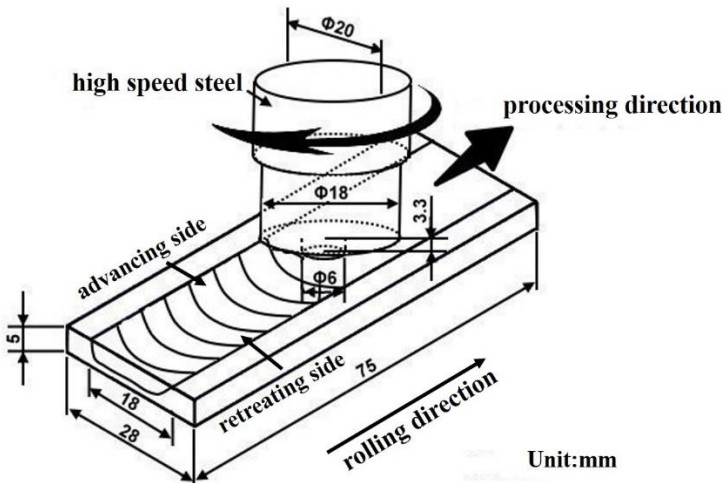

**Figure 1.** Schematic illustration of the friction stir process (FSP).

The sampling position and the dimensions of the tensile specimens are shown in Figure 2. The cross-section area of the tensile specimens with the longitudinal orientation was $4 \times 2$ mm$^2$ and the gauge length was 10 mm (ASTM B 557M). The tensile test was performed with a universal testing machine (HT-8336, Hung Ta, Taichung, Taiwan) at an initial strain rate of $1.67 \times 10^{-3}$ s$^{-1}$ at room temperature. The tensile properties of specimens were tested three times.

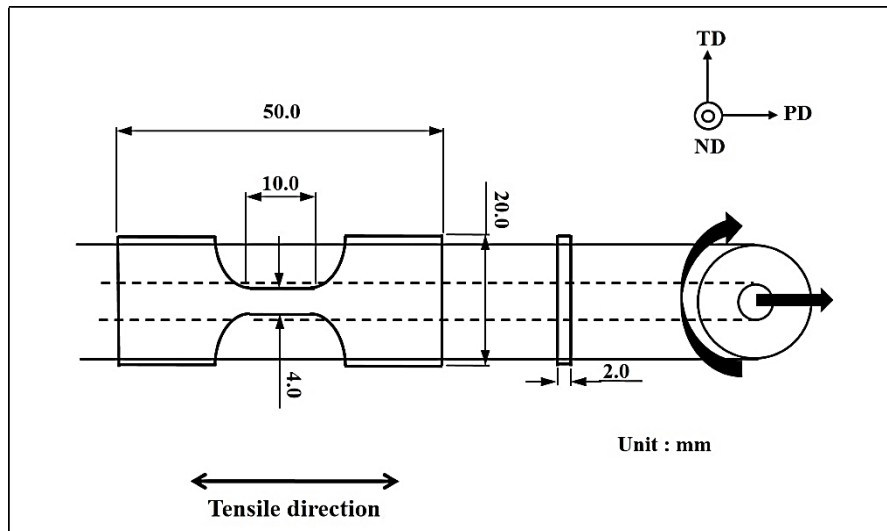

**Figure 2.** Schematic illustration of the tensile specimen (FSP). Processing direction: PD; normal direction: ND; transverse direction: TD.

We observed the microstructure of the 7075-T6 aluminum alloy after FSP and various heat treatments by optical microscope (OM, OLYMPUS BX41M-LED, OLYMPUS, Tokyo, Japan). Meanwhile, the morphology and the chemical composition of the precipitate and the second phase particles were examined by a scanning electron microscope (SEM, HITACHI SU-5000, HITACHI, Tokyo, Japan) equipped with an energy dispersive spectrometer (EDS) in order to characterize the relationship between the precipitate and the second phase particles. X-ray diffraction (XRD, Bruker AXS GmbH, Karlsruhe, Germany) analysis with Cu K$_\alpha$ ($\lambda = 1.541838$ Å) radiation was employed at $2\theta = 20$–$90°$ to identify the intermetallic compounds.

## 3. Results and Discussion

### 3.1. High Improvement of Tensile Ductility in the FSPed Sample via W Treatment

Based on previous reports [27,28], high-strength materials have small work hardening and poor ductility. The 7075 aluminum alloy after a T6 aging treatment has a smaller work hardening than it does after other heat treatments such as full annealing. Hence for the 7075 aluminum alloy, the ductility must be improved and the strength must be decreased before plastic deformation. Table 2 shows the tensile properties of 7075-BM and FSPed specimens after various heat treatments as described in Table 1. The 440-NA specimen had the highest tensile strength because of the effect of FSP and the precipitates (natural aging). However, the ductility of the 440-NA specimen was the lowest among all specimens. In addition, with the W treatment, the tensile strength of the FSPed specimen (440-W) was higher than that of the BM specimen because of the grain refinement. The 440-W specimen indicated the highest tensile elongation (40.3%), it was at least 1.5-fold larger than that of both the annealed BM-O and 440-NA samples (22%). Furthermore, the 440-W specimen's ultimate tensile strength (UTS), indicating onset necking, was also significantly prolonged. Figure 3 compares four examples of tensile deformation curves for the BM and FSPed samples recorded at an identical tensile strain rate. The increments in the tensile elongation of the BM-W and 440-W specimens are attributed to the suppression of the Zn and Mg consumed in forming the Zn- and Mg-rich precipitates.

**Table 2.** The tensile properties of 7075-BM and FSPed specimens after various heat treatments. YS: yield strength; UTS: ultimate tensile strength; UE: uniform elongation; TE: total elongation.

| Specimen Code | YS (MPa) | UTS (MPa) | UE (%) | TE (%) |
| --- | --- | --- | --- | --- |
| BM-O | $102.0 \pm 7.1$ | $208.0 \pm 3.1$ | $13.8 \pm 1.4$ | $25.3 \pm 1.9$ |
| BM-W | $141.0 \pm 4.7$ | $367.0 \pm 6.4$ | $30.1 \pm 2.0$ | $33.4 \pm 0.6$ |
| 440-NA | $288.0 \pm 14.6$ | $454.0 \pm 10.4$ | $17.6 \pm 1.2$ | $22.5 \pm 2.1$ |
| 440-W | $206.0 \pm 9.9$ | $408.0 \pm 8.5$ | $32.5 \pm 2.8$ | $40.3 \pm 0.8$ |

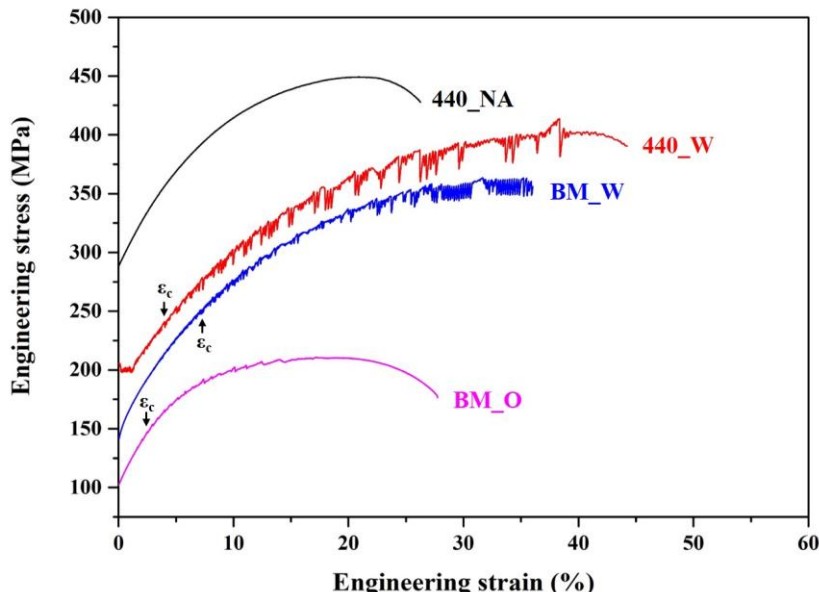

**Figure 3.** Stress–strain flow curves of 7075 aluminum alloy after FSP and various heat treatments.

Compared to the BM and O specimens, the W specimen is typically solutionized at a high temperature (480 °C) in order to obtain a supersaturated solid solution state (SSSS). As is typical of an SSSS sample, the tensile deformation curve of the FSPed specimen (440-W) exhibits two unusual behaviors (Figure 3). The first behavioral phenomenon is the appearance of a stress plateau at the

yield point, which is often a result of interactions between the gliding dislocations and solute atoms and is caused by grain refinement, as is the case with the Piobert–Lüders effect [29–31]. The second phenomenon is the appearance of a serrated flow, where serrated flow instability occurs at a critical point corresponding to a critical strain ($\varepsilon_c$).

Aluminum alloys often exhibit serrated flow after a critical strain during plastic deformation in the range of −50 °C to 100 °C. Zn and Mg atoms are the main factor causing serrated yield [10,32,33]. For the 440-W specimen, serrated flow occurred predominantly in association with higher Zn and Mg concentrations. Compared to the 440-W specimen, the 440-AF specimen (Figure 4) had fewer serrations and the original Zn and Mg concentrations were also lower than those of the 440-W specimen. On the other hand, for the 440-NA specimen, with natural aging at 40 °C for 100 h after the friction stir process, serrations were expected to disappear when the supersaturated Zn and Mg atoms were consumed by aging precipitations.

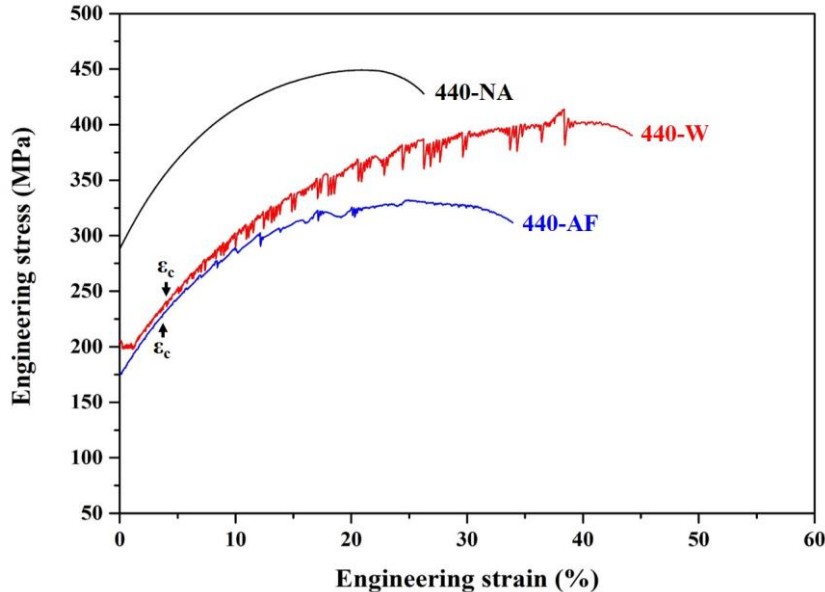

**Figure 4.** Stress–strain flow curve of FSPed specimens with different solid solubility.

As shown in Figures 3 and 4, the critical strain ($\varepsilon_c$) and the stress amplitude of the serrated flow can vary according to the process and is typically associated with the post-heat treatment condition. Many previous reports [10,12–14] have suggested that the Zn and Mg atoms dissolved in the aluminum matrix will act as obstacles to the motion of dislocations and, therefore, will inhibit such motion during tensile deformation. This type of inhibition will manifest in the form of serrated amplitude and the onset of serration strain that can be partly attributed to grain refinement. In addition, the mobile dislocations in finer grain materials encounter obstacles at a higher frequency than in coarser grain materials. According to previous reports [10,23,34–36], after severe plastic deformation, materials have higher initial dislocation density and a higher concentration of vacancies. Hence, the FSPed specimen (440-W specimen) possessed higher initial dislocation density and finer dynamic recrystallization grains than did the BM-W specimen. Compared with the BM-W specimen, the 440-W specimen had a smaller $\varepsilon_c$ and larger stress drop.

For the 440-W specimen, the stress–strain curve showed a plateau at the initial stage just after the end of elastic deformation. In the literature [37], this is called the yield-point phenomenon as well as the Piobert–Lüders effect [29–31]. Tensile deformation is followed by a serrated Lüders extension. The plastic deformation beyond this Lüders strain is characterized by a steady increase of the serration amplitude until the final fracture. Based on reports [38,39], the yield-point phenomenon by Lüders band propagation is strongly affected by grain size. The frequency of serrations also increases with decreasing grain size.

Owing to the severe high-temperature deformation and from the viewpoint of the effects of intermetallic particles and precipitations, 440-W specimens solutionized at 480 °C and rapidly quenched, typically contain very few coarse intermetallic particles (Al-Cu-Fe compound, <10 μm). However, small Zn- and Mg-rich intermetallic particles (<1 μm) actually were dissolved. This microstructural feature will be quantitatively examined later. Figure 5 demonstrates the tensile fractural feature, and all of the specimens exhibited a dimpled pattern, although the fracture patterns differed only slightly. However, the dimpled patterns of the BM-O, BM-W, and 440-W specimens were comparatively larger and deeper than those of the other specimens (Figure 5a–d).

The results of the tensile test indicate that the elongation values of the 440-W and 440-NA specimens were strongly dependent on the ability to prevent necking from occurring at the earlier stage during tensile deformation, as well as on the ability to prevent the generation and accumulation of dislocations, which caused significant strain localization. Fractural features were observed using SEM at high magnification, as shown in Figure 6a,b, where a comparison of the 440-NA specimen and the 440-W specimen indicate that the 440-NA specimen actually had many more Zn- and Mg-rich intermetallic particles and $MgZn_2$ precipitates than did the 440-W specimen. One significant characteristic of the fracture surface as shown in Figure 6a,b, is that an obvious, relatively smooth surface could be observed, which is the typical fracture characteristic of the 440-W specimen as a result of 480 °C solutionizing.

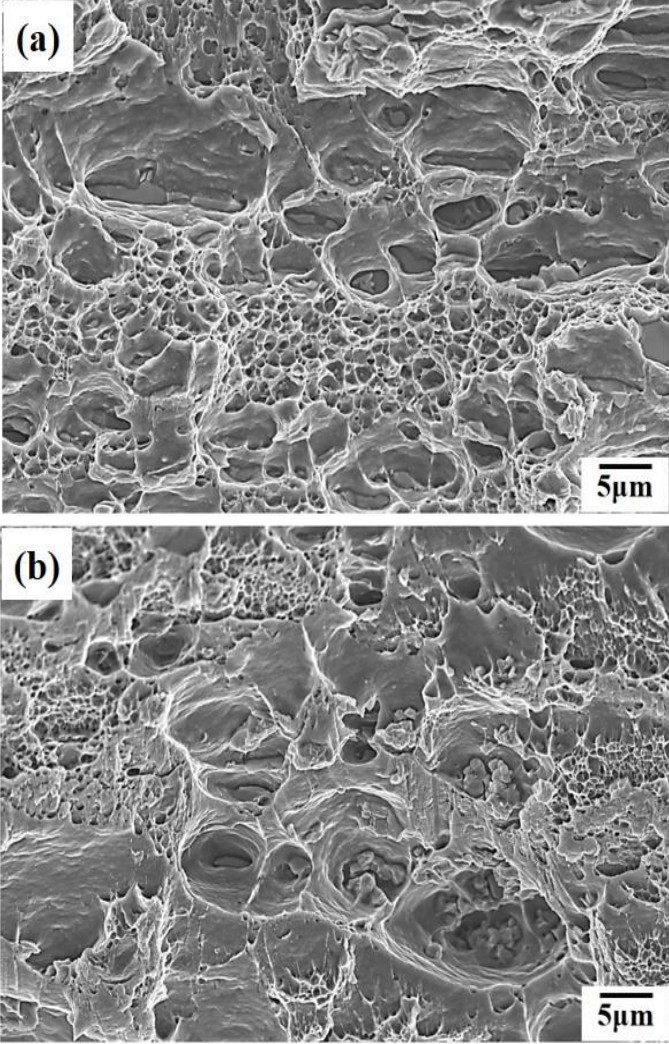

**Figure 5.** *Cont.*

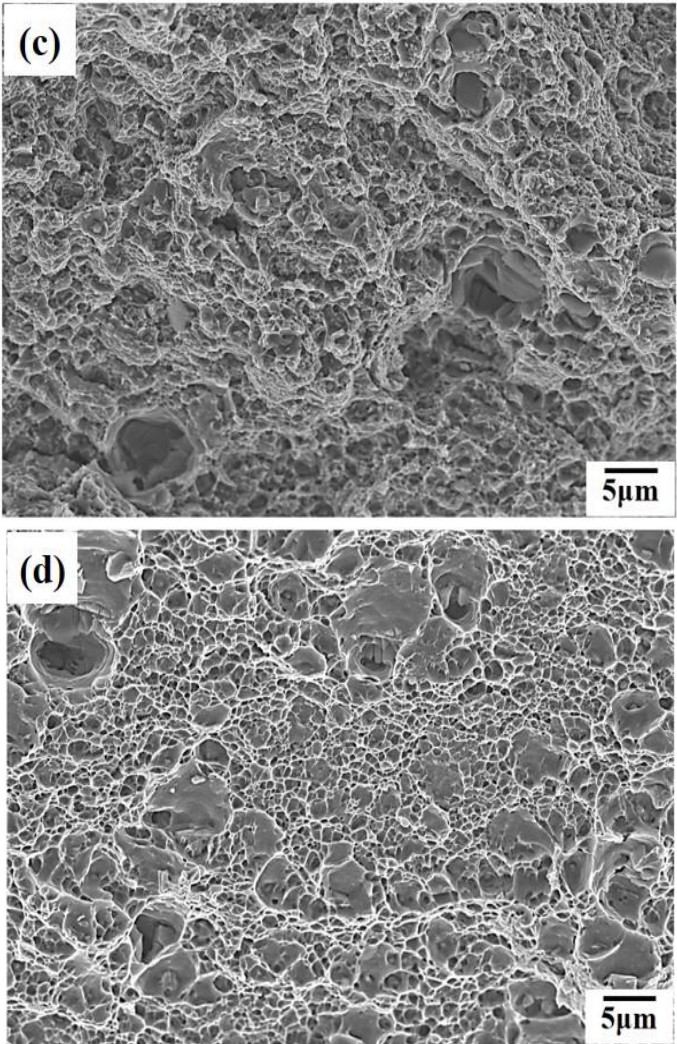

**Figure 5.** Microstructures of tensile fractures for 7075 aluminum alloy after FSP and various heat treatments: (**a**) BM-O specimen, (**b**) BM-W specimen, (**c**) 440-NA specimen, (**d**) 440-W specimen.

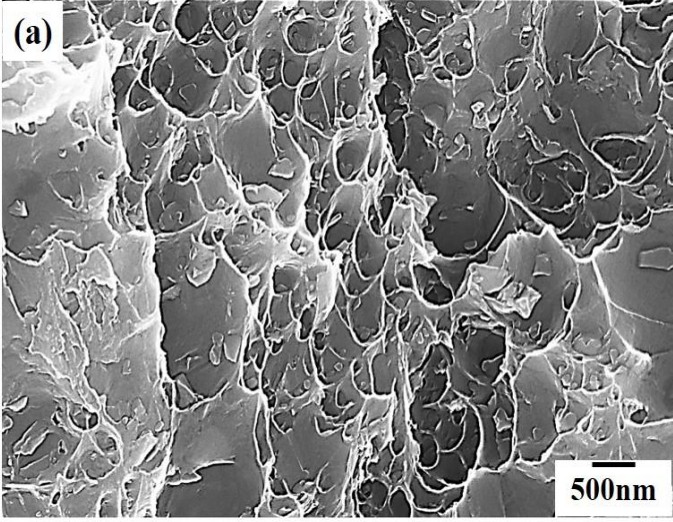

**Figure 6.** *Cont.*

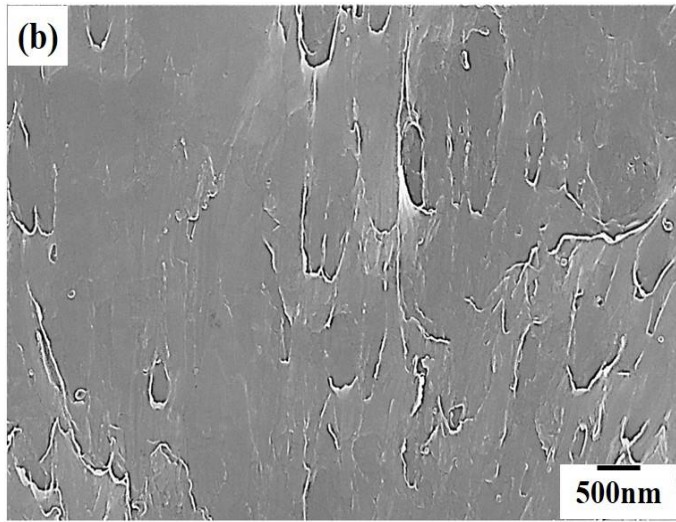

**Figure 6.** SEM images with high magnification of fractures of 440 specimens after various heat treatments (**a**) 440-NA specimen, (**b**) 440-W specimen.

### 3.2. Microstructural Feature of the As-Solutionized and FSPed Materials

As mentioned above, the contribution of the 480 °C solutionizing of the W specimen to the tensile deformation behavior of the AA7075 alloy can be estimated using the critical onset strain ($\varepsilon_c$) and the stress drop of the serrated flow, which are related to either the concentration of solutes or the distribution of the precipitates. Both the critical onset strain ($\varepsilon_c$) and the stress drop tend to increase with solid solubility. On the other hand, estimating by quantifying the amounts of intermetallic particles dissolved in the matrix can also lead to the acquisition of meaningful evidence. Figure 7 presents the peak of the Al matrix, $MgZn_2$, and other intermetallic phases after various heat treatments. Compared with the BM-O and 440-NA specimens, the peak intensity of $MgZn_2$ was weakest after the W treatment, which means many Zn and Mg atoms were dissolved in the matrix. In addition, Figure 7 shows the XRD analysis of the Zn and Mg concentrations. The observed differences can be attributed to the Zn and Mg consumed in forming the Zn- and Mg-rich precipitates.

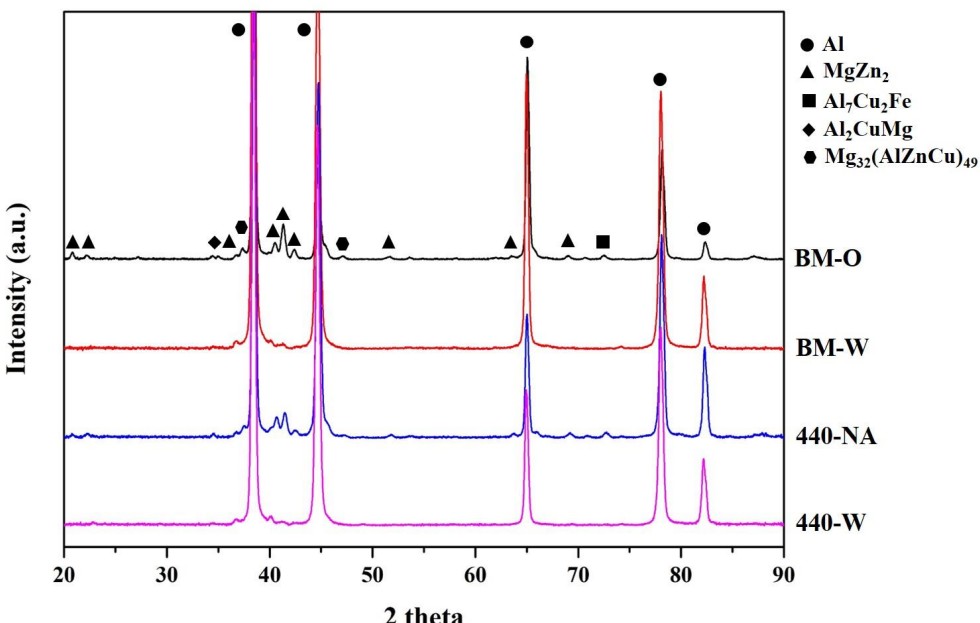

**Figure 7.** The XRD analysis of 7075 aluminum alloy after FSP and various heat treatments.

Figure 8 presents the microstructural feature of the samples after different post-heat treatments. The grain size and the intermetallic particles are shown in Figure 8, where it can be seen that there is little difference in terms of grain size between the BM-O and BM-W samples (Figure 8a,b). The BM-O and BM-W samples had an average grain size of 92 µm and 89 µm, respectively. Figure 8c,d compares examples of the 440-AF sample, where the stir zone (SZ) is characterized by uniformly fine, dynamically recrystallized grains with an average grain size of about 4 µm. The 440-W specimen differed only slightly (6 µm), although it was solutionized at 480 °C. The low grain growth rates of the FSPed 7075 alloy samples indicates development through a continuous dynamic recrystallization (CDRX) mechanism, which could be attributed to thermal mechanical deformation and insufficient exposure duration.

The difference in the morphology and the composition of intermetallic particles examined by the SEM/EDS, as shown in Figure 9 and Table 3, suggest that the intermetallic particles could be divided into the following types: (1) Al-Cu-Fe composition, including a $Al_7Cu_2Fe$ phase (irregular or oval-shaped) in the Al matrix and (2) Al-Mg-Si-Cr phase (irregular or spherical). The Al-Cu-Mg composition, including $Al_2CuMg$ (oval-shaped), was observed at the grain boundary of the specimen after full annealing as indicated in Figure 9a. In addition, the smaller particles (<1 µm) in Figure 9a,c were $MgZn_2$ phase, as determined by our XRD analysis (Figure 7) and the reports [6,40,41]. After 480 °C solution treatment and rapid quenching (W specimen), most of the small intermetallic particles dissolved, and the concentration of Mg and Zn atoms tended to increase in the matrix, as shown in Figure 9b,d.

**Table 3.** EDS data for specimens shown in Figure 9 (Unit: at. %).

| No. | Zn | Mg | Cu | Cr | Si | Fe | Al |
|-----|-----|------|------|-----|------|-----|------|
| A | 0 | 0 | 2.8 | 0 | 0 | 9.7 | 87.5 |
| B | 0 | 24.0 | 24.4 | 0 | 0 | 0 | 51.6 |
| C | 0 | 38.1 | 0 | 6.3 | 23.3 | 0 | 32.3 |
| D | 0 | 0 | 3.5 | 0 | 0 | 8.4 | 88.1 |
| E | 0.9 | 2.8 | 0 | 3.0 | 22.6 | 0 | 70.7 |
| F | 0 | 0 | 19.2 | 0 | 0 | 6.2 | 74.6 |
| G | 1.2 | 7.7 | 0 | 2.1 | 49.8 | 0 | 39.2 |
| H | 0 | 0 | 18.3 | 0 | 0 | 5.4 | 76.3 |
| I | 0.6 | 2.4 | 0 | 4.8 | 66.4 | 0 | 25.8 |

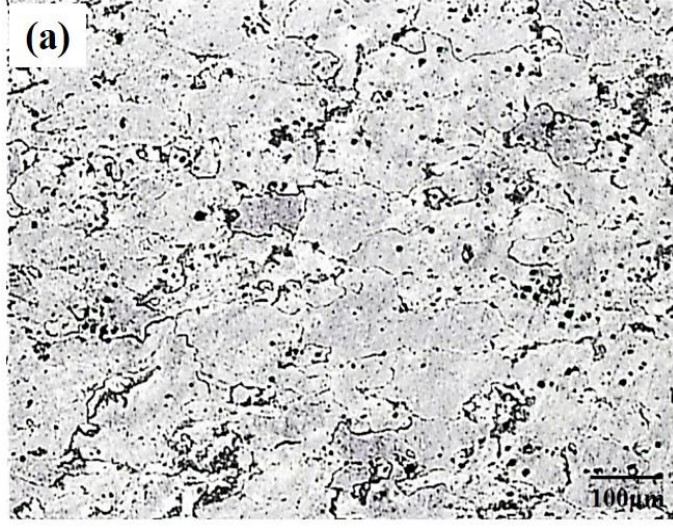

**Figure 8.** *Cont.*

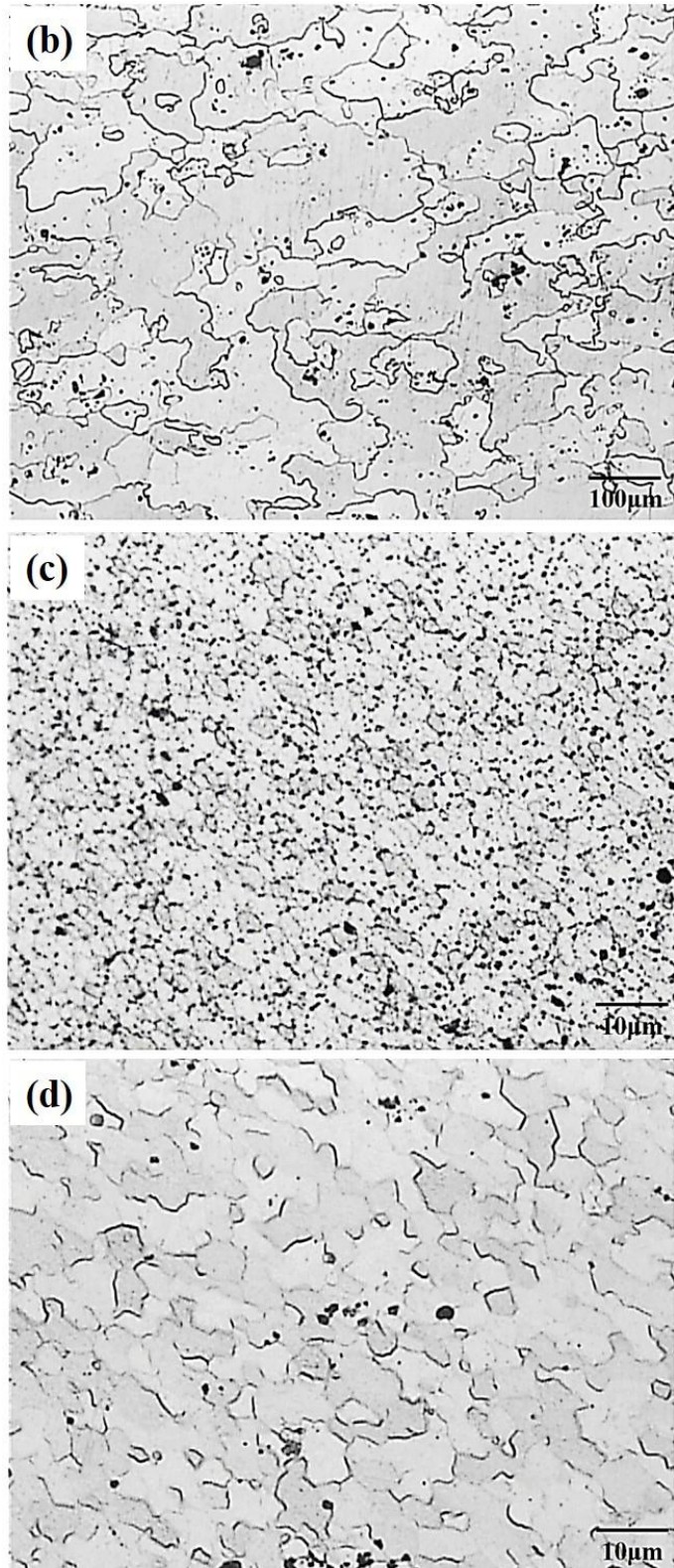

**Figure 8.** The microstructure of 7075 aluminum alloys after FSP and various heat treatments: (**a**) BM-O specimen, (**b**) BM-W specimen (**c**) 440-NA specimen, (**d**) 440-W specimen.

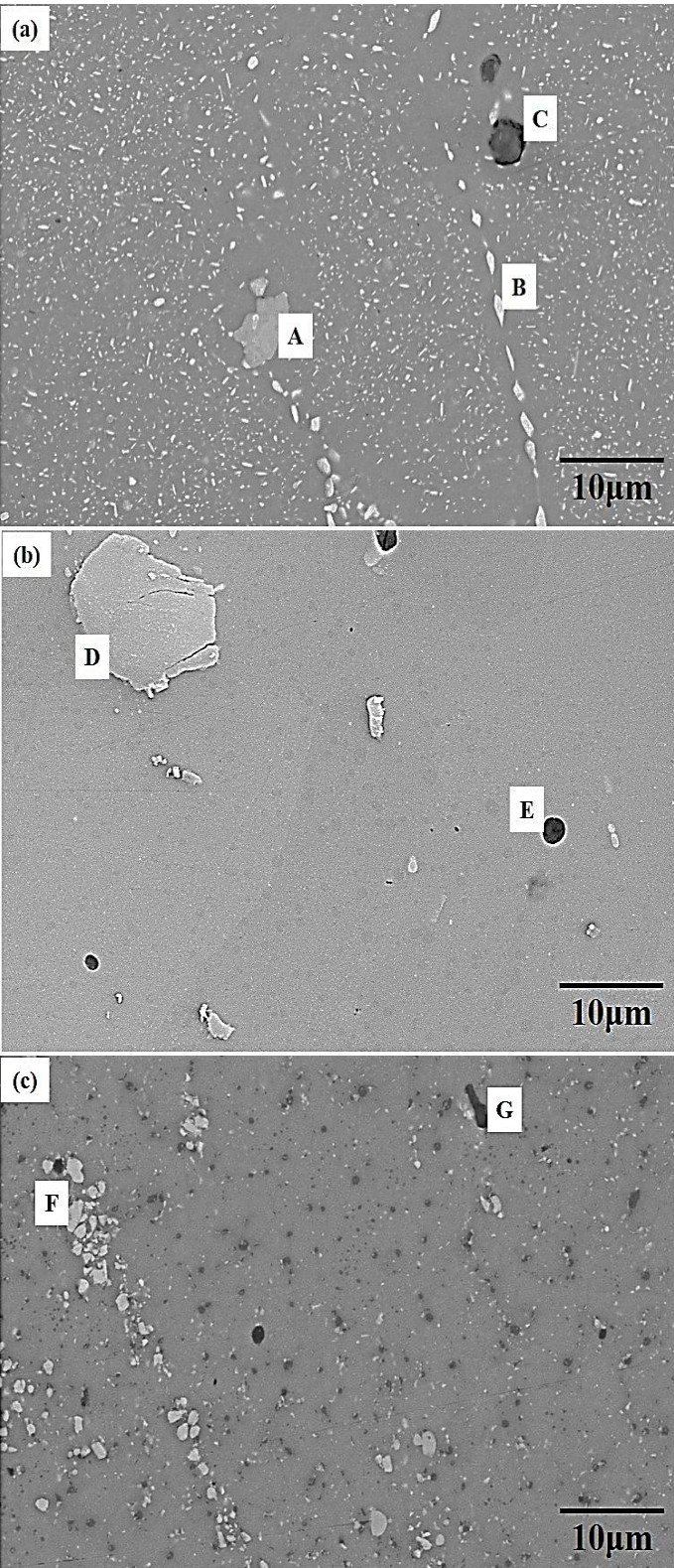

**Figure 9.** *Cont.*

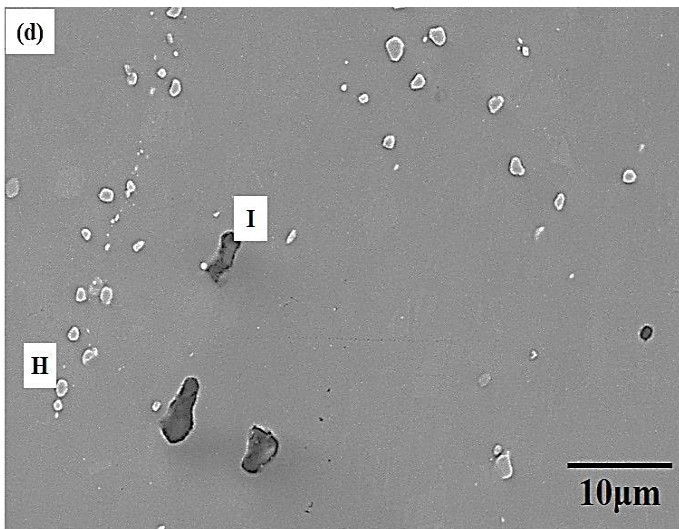

**Figure 9.** The morphology and the distribution of the 7075 aluminum alloys after FSP and various heat treatments: (**a**) BM-O specimen, (**b**) BM-W specimen (**c**) 440-NA specimen, (**d**) 440-W specimen.

### 3.3. Correlation of the Improvement of Tensile Ductility due to the Introduction of W Heat Treatment and FSP

FSPed aluminum alloys have been examined in numerous studies to explore the possibility of inducing previously unobserved microstructural features and mechanical properties. Based on the experimental tensile results mentioned above, the room temperature ductility of high strength 7075 alloys, particularly the tensile elongation of the W and FSPed samples, were improved more than 1.5-fold by FSP followed by W treatment. This could be attributed to the increases in the work-hardening ability, which affected the tensile ductility of the samples.

Figures 3 and 4 exhibited the tensile deformation flow curves of W specimens and indicated that the factors that affected the critical onset strain ($\varepsilon_c$) and the stress drop of the serrated flow included the kind of solute atoms and their solid solubility [42,43], dislocation density [23,34,44], and grain size [22,33,38,39,45]. Sun et al. [11] documented the relationship between solute concentration and the PL effect of the Al-Cu alloy after W treatment at various temperatures. Sato et al. [23] reported that FSPed specimens possessed higher retention of dislocation density and finer grain size, promoting the occurrence of serrations. Based on the results examined in this investigation, the critical onset strain ($\varepsilon_c$) and the stress drop of BM-W and 440-W specimens were larger than those of BM-O specimens. According to previous reports [10–12,42,43], the critical onset strain ($\varepsilon_c$) and the stress drop of the serrated flow are related to either the concentration of solutes or the distribution of precipitates. Both the critical onset strain ($\varepsilon_c$) and the stress drop increased with solid solubility, these effects might act to cause an increase in work-hardening behavior.

The stress–strain relationship of the samples used in this study can be described by Hollomon equation ($\sigma = K_H \, (\varepsilon)^n$). The magnitude of the n value can be determined from the slope of the logarithmic plot of the stress versus strain curve. If the curve is not linear, it indicates a multi-stage work-hardening feature prior to the necking of a tensile specimen. Our previous investigations [25,26,46,47] have confirmed the work-hardening behavior, which can be accommodated prior to the onset of necking; a tensile specimen with a higher n value is capable of absorbing the tensile strain, thus suppressing the strain localization to acquire better uniform elongation.

In general, the n value of an alloy is a compositionally and microstructurally sensitive exponent, so it is appropriate to suggest that part of work-hardening may be acquired via the FSPed fine grains. A remarkable effect due to the FSP was the creation of high angle grain boundary grains (HAGB) resulting from dynamic recrystallization. These HAGBs had few extrinsic dislocations, and were therefore more effective in blocking slipping dislocations. However, it is reasonable to suggest that the HAGB-related deformation activities occurred more easily, especially for the fine grain FSPed

specimens during tensile deformation. Owing to the stress concentrations at the triple junctions of the grain boundary, these effects may act to cause increases in the work-hardening behavior.

The various post-heat treatment conditions discussed above can be used to adjust the solid solution concentration and the number of intermetallic particles or amount of aging precipitation. The effects of such experimental parameters on the tensile deformation behavior and the resultant tensile ductility of the FSPed and W treated AA7075 alloys have not been well examined. The W treated samples indicated that the Zn and Mg existed primarily in the form of a supersaturated solution, where the Zn and Mg atoms dissolved in the matrix constituted point defects that inhibited the motion of mobile dislocations that occurred during tensile deformation, which were manifested as serrations in the tensile flow curve, finally resulting in a higher n value.

The effects of the grain size, precipitates, and heat treatments on the ductility and methods for manufacturing AA7075 alloys that possess high tensile ductility are shown in Figure 10.

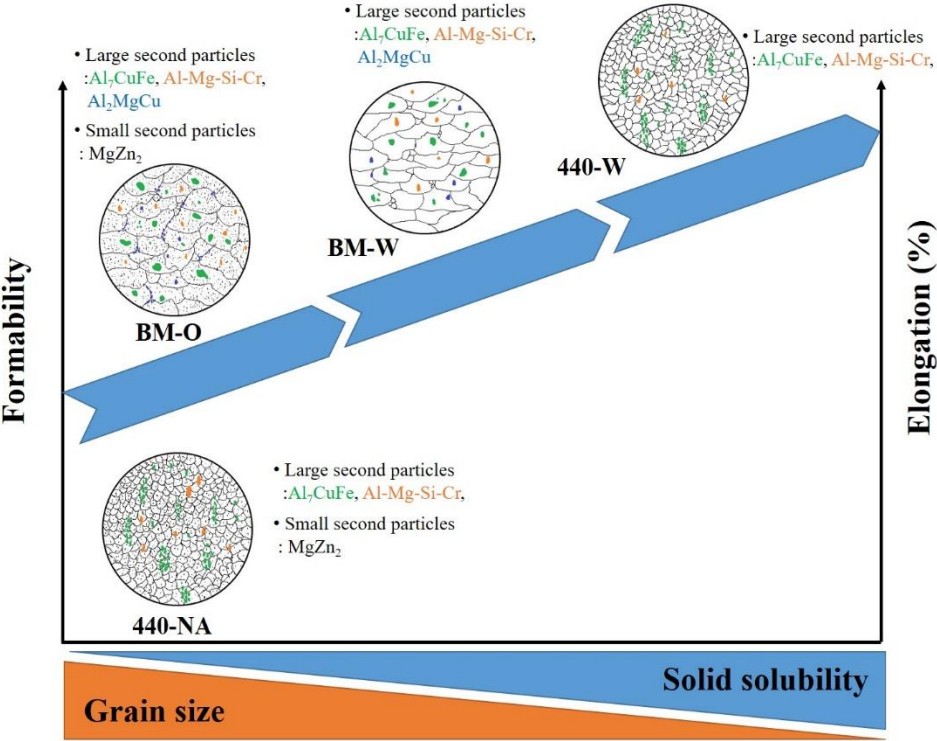

**Figure 10.** A schematic diagram of the relationships among formability, elongation, and microstructure for 7075 aluminum alloys after FSP and various heat treatments.

## 4. Conclusions

A commercial AA7075 aluminum alloy hot rolled plate was subjected to heat treatment, and a friction stir process was used to improve tensile ductility and productivity. The experimental results of this study are as follows:

(1) The ductility of the 7075 aluminum alloy was improved by the FSP and W treatment, where the 440-W specimen exhibited a high improvement in tensile elongation (40.3%) that was at least 1.5-fold greater than that of the annealed BM-O and 440-NA samples (22%). The main cause was investigated, and it was determined that this was correlated with the Zn and Mg consumed during the formation of the Zn- and Mg-rich precipitates and the grain refinement. Furthermore, the ultimate tensile strength (UTS) of the samples indicated that the onset necking was also significantly prolonged.

(2) Most of the small intermetallic particles dissolved after the 480 °C solution treatment and rapid quenching (W specimens), and the concentrations of the Mg and Zn atoms tended to increase

in the matrix. The differences in the morphology and the composition of the intermetallic particles indicated that the intermetallic particles could be divided into (1) Al-Cu-Fe composition, including $Al_7Cu_2Fe$ phase (irregular or oval-shaped) in the Al matrix and (2) Al-Mg-Si-Cr phase (irregular or spherical). In addition, the smaller particles (<1 μm) could be determined as the $MgZn_2$ phase.

(3) The room temperature ductility of the high strength 7075 alloy was improved more than 1.5-fold by the FSP combined with W treatment. This could be attributed to the increases in the work-hardening ability, which affected the tensile ductility of the samples.

(4) The yield-point phenomenon (Piobert–Lüders effect) occurred for the FSPed specimen with W treatment due to grain refinement and high solid solubility, and the Portevin–LeChatelier (PL) effect occurred during the tensile deformation of the W-treated materials. The critical onset strain and the amplitude of the stress drop in the serrations were correlated with the grain size, the precipitates, and the concentrations of solute atoms. The supersaturated Zn and Mg atoms dissolved in the matrix constituted point defects that inhibited the motion of the mobile dislocations occurring during tensile deformation, which manifested as serrations in the tensile flow curve, finally resulting in a higher n value.

**Author Contributions:** M.-H.K. performed the experiments, analyzed the data, and wrote the paper. F.-Y.H. and T.-S.L. are advisers.

**Funding:** This research received no external funding.

**Acknowledgments:** The authors are grateful to The Instrument Center of National Cheng Kung University and the Ministry of Science and Technology of Taiwan (Grant No. MOST 107-2221-E-006-012-MY2) for their financial support for this research.

**Conflicts of Interest:** The authors declare no conflict of interest.

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
