# Peer review of "Examination of the High Tensile Ductility Improvement in an As-Solutionized AA7075 Alloy with the Aid of a Friction Stir Process"

_metals, doi:10.3390/met9020196_

Reviewer 1 Report

The authors present an investigation of the ductility enhancement of a AA7075 alloy induced by friction stir process. The results show that ductility was greatly enhanced by FSP followed by quenching in cool water.

The design of the experiment looks appropriate, and the results are interesting. 

However, the manuscript suffers of a very very low english level. As a rewiever, I am supposed to judge the scientific level of a readable manuscript. I can assure that reading this manuscript was very difficult.

 I recommend the authors to completely revise it, possibly with the help of some english-speaking colleague, or some english service on the web.

Regarding the scientific aspects of the paper, these are my suggestions:

- in lines 78,79, the authors cite the rolling direction (RD). They should indicate such direction in Fig. 1 and/or Fig. 2.

- in line 143, after "deformation" the authors should give some reference.

- Fig. 7 is only discussed in the text in lines 191,192: "Figure 7 reveals that the Mg concentration determined from the XRD analysis". This is an example of what I mentioned previously. What that means? Were the authors give the Mg concentration determined from XRD analysis?

- the "Materials and Methods" section should contain details about the instrumentation used for all the measurements.

After having fixed the above points, and after a deep revision of the english form, the paper could be acceptable for publication.

Author Response

Response to the Reviewer’s Comments

The manuscript suffers of a very low English level. As a reviewer, I am supposed to judge the scientific level of a readable manuscript. I can assure that reading this manuscript was very difficult. I recommend the authors to completely revise it, possibly with the help of some English-speaking colleague, or some English service on the web.

Our response:

We greatly appreciated your mention. The paper has been carefully revised once more by a native English speaker to improve the grammar and readability, including lines 17-19, lines 31-33, lines 43-45, lines 66-72, lines 132-133, lines 135-139, lines 155-156, lines 174-177, lines 181-184, line 221-225, lines 255-256, lines 278-282, lines 289-293, and lines 305-311.

In lines 78-79, the authors cite the rolling direction (RD). They should indicate such direction in Fig. 1 and/or Fig. 2.

Our response:

We greatly appreciated your mention. We amended the Figure 1 and added the arrow of the rolling direction (RD) in line 84.

line 84

Figure 1. Schematic illustration of the friction stir process (FSP).

In line 143, after "deformation" the authors should give some reference.

Our response:

We greatly appreciated your mention. We added some references, including reference [10], [12], [13], [14] in line 153.

[10] Pink, E. The effect of precipitates on characteristics of serrated flow in AlZn5Mg1. Acta Mater. 1989, 37(7), 1773-1781.

[12] Chan, K.S.; Chen, L.H.; Lui, T.S. The effect of particles on the critical strain associated with the Portevin-Le Chatelier effect in aluminum alloys. J. Mater. Sci. 1995, 30, 212-218.

[13] Chen, M.C.; Chen, L.H.; Lui, T.S. Analysis on the critical strain associated with the onset of Portevin-Le Chatelier effect of substitutional F.C.C alloys. Acta. Metall. 1992, 40(9), 2433-2438.

[14] Chen, M.C.; Chen, L.H.; Lui, T.S. A modification on the Portevin-Le Chatelier effect of substitutional FCC alloys. Scr. Metall. 1989, 23, 655-658.

Fig. 7 is only discussed in the text in lines 191,192: "Figure 7 reveals that the Mg concentration determined from the XRD analysis". This is an example of what I mentioned previously. What that means? Were the authors give the Mg concentration determined from XRD analysis?

Our response:

We greatly appreciated your mention. In this study, we just performed the qualitative analysis of all specimens on the phase by XRD and SEM. Hence, we amended the context in this manuscript in lines 209-212. In addition, we also added the description of Zn concentration in this manuscript. 

lines 209-212

Figure 7 provides the Zn and Mg concentrations were compared qualitatively by the XRD analysis. The observed differences can be attributed to the Zn and Mg consumed in forming the Zn- and Mg-rich precipitates.

"Materials and Methods" section should contain details about the instrumentation used for all the measurements.

Our response:

We greatly appreciated your mention. We amended more details about the instrumentation used for all the measurements in lines 93-103.

lines 93-103

The tensile test was performed with a universal testing machine (HT-8336, Hung Ta, Taichung, Taiwan) at an initial strain rate of 1.67×10-3 s-1 at room temperature. The tensile properties of specimens were test three times.

We observed the microstructure of the 7075-T6 aluminum alloy after FSP and various heat treatments by optical microscope (OM, OLYMPUS BX41M-LED, OLYMPUS, Tokyo, Japan). Meanwhile, the morphology and the chemical composition of the precipitate and the second phase particles were examined by a scanning electron microscope (SEM, HITACHI SU-5000, HITACHI, Tokyo, Japan) equipped with an energy dispersive spectrometer (EDS) in order to characterize the relationship between the precipitate and the second phase particles. X-ray diffraction ( XRD, Bruker AXS GmbH, Karlsruhe, Germany) analysis with Cu Kα (λ= 1.541838 Å) radiation was employed from 2θ= 20-90° to identify the intermetallic compounds.

Reviewer 2 Report

The subject discussed in this paper fits the scope of Materials. However, the paper is not publishable in its current form and needs a major revision as detailed below.

First, the title of the paper is inappropriate as the author's data on the improvement of the ductility does not support the claim  of "vast" improvement. Specifically, the authors claimed that the ductility of the 440-W (i.e. 40%) doubles that of the BM-O (i.e., 25.3%) and 440-NA (i.e., 22.5%). This is plainly an inaccurate claim. Also, the ductility comparison was not extended to that of BM-W (i.e., 33.%). Thus providing a false claim.  

Focusing on ductility improvement alone gives a false impression on the overall effects of the processing and heat treatment methods on the mechanical properties of the materials. Detail discussions should have included the effects of these processing methods on other properties such as the yield strength and the ultimate tensile strength of the materials presented in Table 2. Thus, the paper in the present form omits a lot of information.

There are significant grammatical errors and incomplete sentences. For example line 45 and 121 and several others in the text.

The Introduction Section does not provide references to support many claims such as in lines 38-39 where the authors mentioned  "many previous investigations" but failed to cite a single reference. Same in line 65-67.

On line 67, the authors claimed that their previous investigations suggested that they can improve the ductility of aluminum alloys. If this is the case, where are the references and what is the differences between the previous investigations and the current one in this paper?

Statements on lines 68-72 are not clear to the reviewer.

In Section 2, the authors need to provide the details of the solution treatment done in terms of the temperature and time and how these parameters were selected.

How many passes of friction stir was done in this work?

There is no information on the tensile testing machine, the number of samples tested, any variation in the mechanical properties of different samples and the strain acquisition method used.

All the mechanical properties in Table 2 must be discussed and correlated with the materials microstructures

 How was the critical strain in Fig 3 determined?

The claims on line 129 to 135 did not make sense at all. The authors need to show how the concentration of Mg in these processed samples were determined and should also be quantified to make such conclusions. The corresponding micrographs should also be shown. 

The claims on lines 144  to 148 did not make sense at all as the dislocation densities were never quantified. 

Fig 7 was poorly discussed in the text

The reviewer cannot see the average grain sizes mentioned on line 198 

In summary, the paper is not publishable in its current form as it is poorly discussed and presented and lack several important information.

Author Response

Response to the Reviewer’s Comments

First, the title of the paper is inappropriate as the author's data on the improvement of the ductility does not support the claim of "vast" improvement. Specifically, the authors claimed that the ductility of the 440-W (i.e. 40%) doubles that of the BM-O (i.e., 25.3%) and 440-NA (i.e., 22.5%). This is plainly an inaccurate claim. Also, the ductility comparison was not extended to that of BM-W (i.e., 33%). Thus providing a false claim.

Our response:

We greatly appreciated your mention. We amended the title of the paper in lines 2-4. We also more accurately amended the ductility comparison of the 440-W, BM-O, and 440-NA specimens and the value was 1.5-fold above. In addition, we mainly estimated the ductility comparison between the maximum and minimum values of all specimens in this study.

lines 2-4

Examination of the high tensile ductility improvement of as-solutionized AA7075 alloy with the aid of friction stir process

Focusing on ductility improvement alone gives a false impression on the overall effects of the processing and heat treatment methods on the mechanical properties of the materials. Detail discussions should have included the effects of these processing methods on other properties such as the yield strength and the ultimate tensile strength of the materials presented in Table 2. Thus, the paper in the present form omits a lot of information.

Our response:

We greatly appreciated your mention. We amended the context and added the discussions about other tensile properties (YS and UTS) in this manuscript in lines 110-113 and lines 114-118.

lines 110-113

Based on previous report [27, 28], the high strength material have small work hardening and poor ductility. 7075 aluminum alloy after T6 aging treatment has small work hardening than other heat treatment such as fully annealing. Hence, for 7075 aluminum alloy, to improve the ductility and to decrease the strength is needed before plastic deformation.

lines 114-118

The 440-NA specimen had the highest tensile strength because of the effect of FSP and the precipitates (natural aging). However, the ductility of the 440-NA specimen was the lowest among all specimens. In addition, at the W treatment, the tensile strength of the FSPed specimen (440-W) were higher than those of the BM specimen because of the grain refinement.

[27] Cheng, L.M.; Poole, W.J.; Embury, J.D.; Lloyd, D.J. The influence of precipitation on the work-hardening behavior of the aluminum alloys AA6111 and AA7030. Metall. Mater. Trans A. 2003, 34(1), 2473-2481.

[28] Nagarjuna, S.; Srinivas, M.; Balasubramanian, K.; Sarma, D.S. Effect of modulations on yield stress and strain hardening exponent of solution treated Cu-Ti alloys. Scr. Mater. 1998, 38(9), 1469-1474.

There are significant grammatical errors and incomplete sentences. For example line 45 and 121 and several others in the text.

Our response:

We greatly appreciated your mention. The paper has been carefully revised once more by a native English speaker to improve the grammar and readability, including lines 17-19, lines 31-33, lines 43-45, lines 66-72, lines 132-133, lines 135-139, lines 155-156, lines 174-177, lines 181-184, line 221-225, lines 255-256, lines 278-282, lines 289-293, and lines 305-311.

The Introduction Section does not provide references to support many claims such as in lines 38-39 where the authors mentioned "many previous investigations" but failed to cite a single reference. Same in line 65-67.

Our response:

We greatly appreciated your mention. We added some references in line 38 and lines 64-66, including reference [6], [7], [8], [22], [23], [24], [25], [26].

line 38

[6] Ku, M.H.; Hung, F.Y.; Lui, T.S.; Lai, J.C. Enhanced formability and accelerated precipitation behavior of 7075 Al alloy extruded rod by high temperature aging treatment. Metals. 2018, 8(648), 1-14.

[7] Deschamps, A.; Niewczas, M.; Bley, F.; Brechet, Y.; Embury, J.D.; Le Sinq, L.; Livet, F.; Simon, J.P. Low-temperature dynamic precipitation in a supersaturated Al-Zn-Mg alloy and related strain hardening. Philos. Mag. 1999, 79(10), 2485-2504.

[8] Kim, W.J.; Chung, C.S.; Ma, D.S.; Hong, S.I.; Kim, H.K. Optimization of strength and ductility of 2024 Al by equal channel angular pressing (ECAP) and post-ECAP aging. Scr. Metall. 2003, 49, 333-338.

lines 64-66

[22] Lin, C.Y.; Lui, T.S.; Chen, L.H. Microstructural variation and tensile properties of a cast 5083 aluminum plate via friction stir processing. Mater. Trans. 2009, 12, 2801-2807.

[23] Sato, Y.S.; Sugiura, Y.; Shoji, Y. Park, S.H.C.; Kokawa, H.; Ikeda, K. Post-weld formability of friction stir welded Al alloy 5052. Mater. Sci. Eng. A. 2004, 369, 138-143.

[24] Wang, Q.; Zhao, Z.; Zhao, Y.; Yan, K.; Liu, C.; Zhang, H. The strengthening mechanism of spray forming Al-Zn-Mg-Cu alloy by underwater friction stir welding. Mater. Des. 102(2016), 91-99.

[25] Chen, S.T.; Lui, T.S.; Chen, L.H. Effect of revolutionary pitch on the microhardness drop and tensile properties of friction stir processed 1050 aluminum alloy. Mater. Trans. 2009, 8, 1941-1948.

[26] Chen, S.T.; Lui, T.S.; Chen, L.H. Examination of the tensile deformation resistance and ductility of friction stir processes Al-Cu 2218 alloy at elevated temperatures. Mater. Trans. 2010, 8, 1474-1480.

On line 67, the authors claimed that their previous investigations suggested that they can improve the ductility of aluminum alloys. If this is the case, where are the references and what is the differences between the previous investigations and the current one in this paper?

Our response:

We greatly appreciated your mention. Our previous reports [22]; [25; [26] investigated the effect of friction stir process (FSP) before and after post heat treatment such as fully annealing on the microstructure features and tensile properties. For example, the ductility of a cast 5083 aluminum plate [22] via FSP was improved (TE: 12% à 26%) due to the recrystallized fine grain and an intense breakup of the intermetallic compounds. In addition, many previous reports [A-C] also investigated 7075 aluminum alloy after FSP and post heat treatment on the microstructure feature and tensile properties, but few studies investigated the improvement ductility issue. In this study, for 7075 aluminum alloy, we used FSP and W treatment and expected to obtain a higher ductility as well as improve the formability of 7075 aluminum alloy.

[22] Lin, C.Y.; Lui, T.S.; Chen, L.H. Microstructural variation and tensile properties of a cast 5083 aluminum plate via friction stir processing. Mater. Trans. 2009, 12, 2801-2807.

[25] Chen, S.T.; Lui, T.S.; Chen, L.H. Effect of revolutionary pitch on the microhardness drop and tensile properties of friction stir processed 1050 aluminum alloy. Mater. Trans. 2009, 8, 1941-1948.

[26] Chen, S.T.; Lui, T.S.; Chen, L.H. Examination of the tensile deformation resistance and ductility of friction stir processes Al-Cu 2218 alloy at elevated temperatures. Mater. Trans. 2010, 8, 1474-1480.

[A] Rajakumar, S.; Muralidharan, C.; Balasubramanian, V. Influence of friction stir welding process and tool parameters on strength properties of AA7075-T6 aluminium alloy joints. Mater. Des. 2011, 32, 535-549.

[B] Mahoney, M.W.; Rhodes, C.G.; Flintoff, J.G.; Spurling, R.A.; Bingel, W.H. Properties of Friction-Stir-Welded 7075 T651 Aluminum. Metall. Mater. Trans. A 1998, 29A, 1955-1964.

[C] Ku, M.H.; Hung, F.Y.; Lui, T.S.; Chen, L.H. Embrittlement Mechanism on Tensile Fracture of 7075 Al Alloy with Friction Stir Process (FSP). Mater Trans. 2011, 52(1), 112-117.

Statements on lines 68-72 are not clear to the reviewer.

Our response:

We greatly appreciated your mention. We amended the context in this manuscript in lines 66-72.

lines 66-72

Our previous investigations [22, 25, 26] suggested that the tensile ductility can be improved for many aluminum alloys through the development of a controlled cooling method for the purpose to acquire the effect of grain refinement, phase transformation or supersaturated solid solution. However, this research focused on the effect of a rapid quenching treatment combined with a friction stir process on the ability to achieve higher tensile ductility. Efforts have been directed toward understanding the deformation kinetics of alloys, which have been predominantly attributed to the solution effects associated with the presence of intermetallic particles in the matrix.

In Section 2, the authors need to provide the details of the solution treatment done in terms of the temperature and time and how these parameters were selected.

Our response:

We greatly appreciated your mention. The processes of all the heat treatment was shown in table 1 in line 85. According to ASM-Metals Handbooks( 9th ed.) and previous reports [4-6], for 7075 aluminum alloy, the solution temperature was form 466℃ to 482℃ depending on product and the solution time was 1 hour above. Hence, we selected the parameters of the solution treatment were at 480℃ for 1 hour and rapidly quenching water. In addition, for the full anneal treatment, we selected three steps: firstly, the sample was at 415 for 2 hours in the furnace, and then allowing the sample to slowly cool ( 20/h cooling rate) to 260, and subsequently at 260 holding 6 hours inside the furnace, and allowing the sample slowly cool to room temperature inside the furnace.

line 85

Table 1. Experimental conditions and specimen codes.

Rotational   speed (rpm)

AF*a

NA*b

W*c

O*d

BM-T6

--

--

--

BM-W

BM-O

440

440-AF

440-NA

440-W

--

*a AF: As fabricated.

*b NA: natural aging (40, 100h).

*c W: solution treatment (480, 1h) + quenching (cool water) + placed in dry ice.

*d O: Heating (415, 2h) + Furnace cooling (20/h) to 260 + heating (260, 6h) + Furnace cooling to room temperature (RT)

ASM-Metals Handbooks: Properties of Wrought Aluminum; 9th ed.; 129-131.

[4]   Sessler, J.; Welss, V. Metallurgy. Materials Data Handbook: Aluminum Alloy 7075, 1st ed.; National Aeronautics and Space Administration: Alabama, USA. 1967; Chap. 3, 9-23.

[5]   Hatch, J.E. Aluminum: Properties and Physical Metallurgy, 1st ed.; American Society for Metals: Ohio, USA. 1984; 152-153; 0871701766, 9780871701763.

[6]   Ku, M.H.; Hung, F.Y.; Lui, T.S.; Lai, J.C. Enhanced formability and accelerated precipitation behavior of 7075 Al alloy extruded rod by high temperature aging treatment. Metals. 2018, 8(648), 1-14.

How many passes of friction stir was done in this work?

Our response:

We greatly appreciated your mention. For the specimens used in this study, we didn’t investigate the effect of overlapping FSP processed sample, the width of one pass sample has already sufficient to acquire an optimum tensile specimen as indicated in Figure 2 (line 106).

line 106

Figure 2. Schematic illustration of the tensile specimen (FSP). (processing direction: PD; normal direction: ND; transverse direction: TD).

There is no information on the tensile testing machine, the number of samples tested, any variation in the mechanical properties of different samples and the strain acquisition method used.

Our response:

We greatly appreciated your mention. We added the information about the tensile testing machine and the number of samples tested in lines 93-95. We also added the standard deviation data of for three specimens in Table 2 in line 126. In addition, we used a calibrated distance between two marks on the specimen surface as an original gauge length (l0) and the distance of the specimen after rupturing as a gauge length (l), the calculated value is the tensile elongation of the specimens as shown in Figure A. In addition, the variation of tensile deformation vs elongation curve were recorded as shown in Figure 3 (line 130) and Figure 4 (line 150)in this manuscript.

lines 93-95

The tensile test was performed with a universal testing machine (HT-8336, Hung Ta, Taichung, Taiwan) at an initial strain rate of 1.67×10-3 s-1 at room temperature. The tensile properties of specimens were test three times.

line126

Table 2. The tensile properties of 7075-BM and FSPed specimens after various heat treatments. (YS: yield strength; UTS: ultimate tensile strength; UE: uniform elongation; TE: total elongation).

YS (MPa)

UTS (MPa)

UE (%)

TE (%)

BM-O

102.0±7.1

208.0±3.1

13.8±1.4

25.3±1.9

BM-W

141.0±4.7

367.0±6.4

30.1±2.0

33.4±0.6

440-NA

288.0±14.6

454.0±10.4

17.6±1.2

22.5±2.1

440-W

206.0±9.9

408.0±8.5

32.5±2.8

40.3±0.8

Figure A.

line 130

Figure 3. Stress-strain flow curves of 7075 aluminum alloy after FSP and various heat treatments.

line 150

Figure 4. Stress-strain flow curve of FSPed specimens with different solid solubility.

All the mechanical properties in Table 2 must be discussed and correlated with the materials microstructures.

Our response:

We greatly appreciated your mention. We amended the context in this manuscript in lines 114-118.

lines 114-118

The 440-NA specimen had the highest tensile strength because of the effect of FSP and the precipitates (natural aging). However, the ductility of the 440-NA specimen was the lowest among all specimens. In addition, at the W treatment, the tensile strength of the FSPed specimen (440-W) were higher than those of the BM specimen because of the grain refinement.

How was the critical strain in Fig 3 determined?

Our response:

We greatly appreciated your mention. According to the reports [A]; [B], the critical strain was the minimum strain needed for the onset of serrations in the stress-strain curve. In the study, the onset strain of the serration is not clear but still could be determined comprehensibly.

[A] Cottrell, A.H. A note on the Portevin-LeChatelier effect, Phi. Mag. 1953, 44, 829-832.

[B] Cottrell, A.H.; Jaswon, M.A. Distribution of solute atoms round a slow dislocation. P. Roy. Soc. Lond, A Mat. 1949, 199, 104-114. 

The claims on line 129 to 135 did not make sense at all. The authors need to show how the concentration of Mg in these processed samples were determined and should also be quantified to make such conclusions. The corresponding micrographs should also be shown.

Our response:

We greatly appreciated your mention. In this study, we just performed the qualitative analysis of all specimens on the phase by XRD and SEM. Hence, we amended the context in this manuscript in lines 209-212. In addition, we also added the description of Zn concentration in this manuscript. 

lines 209-212

Figure 7 provides the Zn and Mg concentrations were compared qualitatively by the XRD analysis. The observed differences can be attributed to the Zn and Mg consumed in forming the Zn- and Mg-rich precipitates.

The claims on lines 144 to 148 did not make sense at all as the dislocation densities were never quantified.

Our response:

We greatly appreciated your mention. We amended the context and added some references [10]; [23]; [34]; [35]; [36] in this manuscript in lines 158-162.

lines 158-162

According to previous repot [10, 23, 34-36] after severe plastic deformation, materials have higher initial dislocation density and higher concentration of vacancy. Hence, the FSPed specimen (440-W specimen) possessed higher initial dislocation density and finer dynamic recrystallization grain than the BM-W specimen. Compared with the BM-W specimen, the 440-W specimen had smaller εc and larger stress drop.

[10] Pink, E. The effect of precipitates on characteristics of serrated flow in AlZn5Mg1. Acta Mater. 1989, 37(7), 1773-1781.

[23] Sato, Y.S.; Sugiura, Y.; Shoji, Y. Park, S.H.C.; Kokawa, H.; Ikeda, K. Post-weld formability of friction stir welded Al alloy 5052. Mater. Sci. Eng. A. 2004, 369, 138-143.

[34] Saha, G.G.;Mccormick, P.G.; Rama Rao, P. Portevin-Le Chatelier effect in an Al-Mn alloy : serration characteristics. Mater. Sci. Eng. 1984, 62, 187-196.

[35] Chen, M.C.; Chen, L.H.; Lui, T.S. Vacancy concentration in strain ageing of substitutional fcc alloys. J. Mater. Sci. 1993, 28, 3329-3334.

[36] Wen, W.; Morris, J.G. The effect of cold rolling and annealing on the serrated yielding phenomenon of AA5182 aluminum alloy. Mater. Sci. Eng. A. 2004, 373, 204-216.

Fig 7 was poorly discussed in the text.

Our response:

We greatly appreciated your mention. We amended the context in this manuscript, including lines 206-212.

lines 206-212

Figure 7 presents the peak of Al matrix, MgZn2 and other intermetallic phase after various heat treatments. Compared with the BM-O and 440-NA specimens, the peak intensity of MgZn2 were weakness after W treatment, which means many Zn and Mg atoms were dissolved in the matrix. In addition, Figure 7 provides the Zn and Mg concentrations were compared qualitatively by the XRD analysis. The observed differences can be attributed to the Zn and Mg consumed in forming the Zn- and Mg-rich precipitates.

The reviewer cannot see the average grain sizes mentioned on line 198.

Our response:

We greatly appreciated your mention. We added the average grain sizes of BM-O and BM-W specimens in lines 217-219.

lines 217-219

The grain size and the intermetallic particle are shown in Figure 8, where it can be seen that there is little difference in term of grain sizes between the BM-O and BM-W samples (Figures 8(a) and 8(b)). The BM-O and BM-W samples had average grain sizes of 92μm and 89μm, respectively.

Round  2

Reviewer 1 Report

The manuscript is now suitable for publication.

Reviewer 2 Report

The authors have addressed the reviewer's concerns. The manuscript can now be published. It is suggested that the authors do thorough proofreading of the manuscript as it still contains some grammatical errors.